# Psychometric validation of the Hospital Stress Questionnaire

**Daniel M. Ford**[1,2], **Rebecca Lawton**[1,2], **Elizabeth A. Teale**[3], **Daryl B. O'Connor**[1*]

**1** School of Psychology, Faculty of Medicine and Health, University of Leeds, Leeds, United Kingdom, **2** Quality and Safety Research Group, Bradford Institute for Health Research, Bradford Royal Infirmary, Bradford, United Kingdom, **3** Academic Unit for Aging and Stroke Research, University of Leeds, Leeds, United Kingdom

\* D.B.OConnor@leeds.ac.uk

## Abstract

### Background

Psychological stress experienced by inpatients has been shown to be associated with poorer post-hospital outcomes. Research that explores and intervenes to address in-hospital stress and ameliorate negative patient outcomes requires a valid measurement tool. The Hospital Stress Questionnaire (HSQ) was developed for this purpose. The aims of the current study were to psychometrically validate the HSQ, identify latent factors, reduce the number of items within the HSQ, and explore the psychometric properties of longer and shorter versions of the scale.

### Methods

A nationally representative sample of recent NHS hospital inpatients (N = 660; mean age = 54.0, range = 18–97) completed the HSQ within a survey of patient experiences; 32 of which completed the measure a second time two weeks later. Factor structure, convergent validity, known-groups validity, predictive validity, and test-retest reliability were assessed.

### Results

Seven domains of in-hospital stress were identified: quality of care, away from home, inconvenienced, health anxiety, negative effects of treatment, ward environment, and disrupted patient experience. Long (55 items), medium (28 items), and short (10 items) versions of the measure were produced, all exhibiting excellent psychometric properties. The highest rated stressor was "poor sleep".

### Conclusion

The HSQ is a valid and reliable tool, now available to be used by researchers and clinicians. It has potential to be used in intervention studies to reduce in-hospital stress, and to identify patients most at risk of the effects of post-hospital syndrome.

**Data availability statement:** All data and R code are accessible here: https://github.com/DMFord97/Validation.

**Funding:** This research was funded by the National Institute for Health and Care Research (NIHR) Yorkshire and Humber Patient Safety Research Collaboration (NIHR Yorkshire and Humber PSRC) and the NIHR Yorkshire and Humber Applied Research Collaboration (NIHR Yorkshire and Humber ARC). The views expressed in this article are those of the authors and not necessarily those of the NIHR, or the Department of Health and Social Care.

**Competing interests:** The authors have declared that no competing interests exist.

# 1. Introduction

## Background

While in hospital, patients are exposed to an abundance of hospital-related stressors, including those relating to physical, psychological, interpersonal, and environmental factors [1]. For example, Goldwater et al [2] identified a number of physical stressors, such as sleep, malnourishment, dehydration, mobility, and pain; while Detsky and Krumholz [3] have proposed several psychological stressors, such as depersonalisation and uncertainty. Interpersonal stressors concern the patient's relationship with hospital staff and other patients [e.g., 4], and environmental stressors focus on aspects of the hospital room or ward, such as lighting and temperature [e.g., 5].

Throughout the hospital stay, the psychological and physiological strain experienced in response to these stressors can lead the body into a state of allostatic overload [6], causing deleterious effects on the patient's health and wellbeing [7,8]. In their review, Guidi and colleagues [7] highlighted that allostatic load has been linked to an increased risk for cardiovascular disease, high systolic and diastolic blood pressure, and depressive symptoms. These effects may then follow the patient home, impeding their recovery, and potentially resulting in an unplanned readmission [9,10] or other poor post-hospital outcomes [11].

Taken together, the above series of events describe a phenomenon known as post-hospital syndrome [12]: an acquired period of generalised vulnerability to adverse events (e.g., post-operative wound infection) following hospitalisation. A growing post-hospital syndrome literature is emerging that outlines associations between hospital-related stressors and patient outcomes [11,13]. For example, Rawal et al. [13] showed that inpatients reporting more disturbances in sleep, mobility, nutrition, and mood – known hospital-related stressors [2] – had a significantly greater risk of unplanned readmission or emergency department visit than their less affected counterparts. This literature presents a clear need to reduce patients' exposure to in-hospital stressors, with the hopes of improving patient outcomes and saving health authorities both money and resources. However, in order to do this, we must first identify, understand, and measure the hospital-related stressors in question.

## The Hospital Stress Questionnaire

The Hospital Stress Questionnaire (HSQ [14]) is a newly developed self-report tool to measure the perceived psychological stress of inpatients in UK hospitals. It identifies 67 hospital-related stressors and allows respondents to rate how stressful they perceived each stressor to be during their hospital stay. The HSQ items were informed by interviewing recent hospital inpatients, who were asked what they found stressful about their hospital stay. The questionnaire builds on previous attempts to measure in-hospital stress, which are often outdated, do not allow for individual differences in perceived stress [15], or are specific to certain populations (e.g., older adults [16,17]).

## Aims of the current study

In its current form, the HSQ has not yet been shown to be valid and reliable, and the length of the questionnaire may be burdensome for patients. Therefore, the aims of the current study were to: (i) test the psychometric properties of the HSQ, (ii) reduce the number of items, (iii) group the remaining items into factors, and (iv) produce medium and short versions of the measure.

# 2. Materials and methods

## Design

A cross-sectional online survey was conducted from March to December 2023. Within psychometric theory, a subject-to-item ratio of 10:1 is recommended for conducting an

exploratory factor analysis [18,19], therefore, we aimed to recruit a sample size of 670 participants. This study received ethical approval from the University of Leeds, School of Psychology Research Ethics Committee (PSYC-774), and was preregistered (AsPredicted #153763).

## Participants

Six hundred and seventy-two completed responses were received, 12 of which were excluded for not meeting the below criteria, leaving a total of 660 participants. Inclusion criteria were the same as for the initial validation of the HSQ (see Ford et al. [14] for justifications), which were as follows: (i) participants were required to be at least 18 years old, (ii) have stayed in a UK hospital as an inpatient, (iii) in the past 12 months, (iv) for at least 24 hours, (v) not for paediatric, maternity, or psychiatric care. A consultee was permitted to assist with or complete the survey on behalf of a relative/friend that was unable to participate themselves.

Participants were recruited between 21st March 2023 and 15th November 2023 from a variety of sources: Prolific (www.prolific.com), Care Opinion (www.careopinion.org.uk), the University of Leeds, School of Psychology Successful Ageing Panel, social media, and word of mouth. All participants provided written informed consent. Those participating via Prolific were compensated with £2 for completion of the study; those recruited via other methods were eligible to be entered into a prize draw to win a £100 gift voucher, or one of three £50 gift vouchers. From the 660 eligible responses, 32 participants were invited to complete the HSQ again, to assess test-retest reliability. For an intraclass correlation coefficient (ICC), a minimum sample of 30 participants has been recommended [20]. These participants were recruited via Prolific, 14 days after their initial response, and were compensated with a further £1.50.

As this study recruited primarily online, several measures were taken to identify bots (automated software misrepresenting as participants [21]) and fraudulent responses – submissions made with fictional data, in an attempt to receive payment for participation (see [22]). To identify such responses, screening questions were placed at the beginning of the survey, three attention checks were added within the survey (e.g., "Please select '7' to show you are paying attention"), and participants were asked for the name of the hospital at which they were admitted. Should a participant fail any screening or attention question, or name a hospital not based in the UK, that response was excluded.

## Measures

The survey was conducted online and took approximately 15 minutes to complete via Qualtrics software (2023). Questions focused on the participant's most recent hospital experience, and began with five screening questions to assess the respondent's eligibility to participate. These were followed by demographic questions including the participant's age, gender, ethnicity, level of education, and marital status. The survey then moved onto hospital-related information such as when their most recent hospital stay was, how long they were in hospital for, which hospital they stayed in, whether or not they had surgery, and whether their stay was planned or an emergency. The survey then presented the following questionnaires: the HSQ (with three attention checks), the Perceived Stress Scale (PSS), and the EuroQol 5-Dimension Health Questionnaire (EQ-5D). See Appendix S1 in S1 File for the full survey.

**The Hospital Stress Questionnaire.** The HSQ [14] is a self-report measure of inpatient psychological stressors, consisting of 67 items measured between 1 (no stress) and 10 (extreme stress). The HSQ has been piloted on 10 laypersons, confirming face validity, and presented to the Yorkshire Quality and Safety Research Group, confirming content validity. Additionally, the measure was completed by 200 persons who had been in hospital in the past 12 months

for at least 24 hours, in order to provide initial validation: both internal consistency ($\alpha = 0.97$) and convergent validity ($r = 0.77$ with PSS-10) were excellent.

**Perceived Stress Scale.** The PSS was included to assess convergent validity of the HSQ. Each item on the scale was reworded from "In the last month…" to "While in hospital…" The PSS-10 [23] was chosen as it is one of the most widely used measures of psychological stress, and has consistently been shown to be reliable and valid [24]. The 10-item version was chosen over the 14-item version due to its shorter length, and chosen over the 4-item version due to its superior internal reliability (Cronbach's alpha: 0.78 vs 0.60 [25]).

**EuroQol 5-dimension health questionnaire.** The EQ-5D was included to assess the predictive validity of the HSQ. It is a widely-used measure for describing and valuing health, composed of five dimensions: Mobility, Self-Care, Usual Activities, Pain/Discomfort, and Anxiety/Depression [26]. Each of these dimensions can be measured using a three-level (EQ-5D-3L) or five-level (EQ-5D-5L) Likert scale [27], both of which have excellent psychometric properties but the five-level version is more sensitive to change [28]. A participant's responses to the EQ-5D reveal their *health state* (e.g., 11111 is indicative of full health); a formula can be applied to this 5-digit code to derive an *index value*, which reflects how good or bad a health state is according to the preferences of the general population of a country/region. These preferences are determined using a *value set*; a representative sample from that country/region – the value set for England was used in the current study. The EQ-5D also includes a visual analogue scale (the EQ VAS): a thermometer where the respondent can indicate self-rated health between 0 (worst possible health) and 100 (best possible health).

## Analysis

Data was analysed using R Statistical Software (v4.3.1) (data and code can be accessed at: https://github.com/DMFord97/Validation). Descriptive statistics were conducted in order to assess the appropriateness of the sample. To identify the latent factor structure of the measure, exploratory factor analysis (EFA) was employed following Watkins' [29] guide to best practice. Bartlett's test of sphericity [30] and the KMO statistic [31] were used to evaluate the suitability of EFA. To determine the number of factors to retain, parallel analysis [32], minimum average partial method [33], visual scree test [34], and Kaiser's criterion [35] were consulted. The oblimin rotation [36] was chosen as it was assumed that factors would be correlated. Due to the large sample size, factor loadings above 0.30 were retained [37].

Internal consistency reliability was assessed using Cronbach's alpha, it has been suggested that acceptable values range between 0.70 and 0.95 [38]. Convergent validity and predictive validity were assessed using Pearson's correlations, where $r = 0.10$, 0.30, and 0.50 were considered small, medium, and large, respectively [39]. Test-retest reliability was assessed using intraclass correlation coefficients (ICC); values less than 0.5, between 0.5 and 0.75, between 0.75 and 0.9, and greater than 0.90 are indicative of poor, moderate, good, and excellent reliability, respectively [20]. Known-groups validity was assessed using an independent samples t-test, exploring mean differences in hospital stress between patients with planned and emergency admissions.

## 3. Results

### Descriptive statistics

To assess the representativeness of our sample, demographics of participants that completed the survey were compared against the latest NHS hospital admission data [40]. Table 1 shows the NHS figures for age, sex, and ethnicity, desired figures adjusted for a sample size of 670,

**Table 1. Demographic data for NHS 2022–23 hospital admissions, with comparisons.**

| Demographic | NHS 2022–23 data | | Desired sample | | Actual sample | |
|---|---|---|---|---|---|---|
| | *n* | % | *n* | % | *n* | % |
| **Age groups** | | | | | | |
| 18–29 | 1,434,124* | 8.2 | 55 | 8.2 | 79 | 12.0 |
| 30–39 | 1,923,957 | 11.0 | 73 | 11.0 | 107 | 16.2 |
| 40–49 | 1,571,431 | 9.0 | 60 | 9.0 | 65 | 9.8 |
| 50–59 | 2,372,014 | 13.5 | 91 | 13.5 | 139 | 21.2 |
| 60–69 | 2,966,772 | 16.9 | 113 | 16.9 | 113 | 17.1 |
| 70–79 | 3,756,596 | 21.4 | 144 | 21.4 | 140 | 21.2 |
| 80–89 | 2,747,191 | 15.7 | 105 | 15.7 | 15 | 2.3 |
| >90 | 760,619 | 4.3 | 29 | 4.3 | 2 | 0.3 |
| Total | 17,532,704 | 100 | 670 | 100 | 660 | 100 |
| **Sex** | | | | | | |
| Female | 9,734,922 | 55.5 | 372 | 55.5 | 366 | 55.5 |
| Male | 7,797,782 | 44.5 | 298 | 44.5 | 292 | 44.2 |
| Other | – | – | – | – | 1 | 0.2 |
| PNS | – | – | – | – | 1 | 0.2 |
| Total | 17,532,704 | 100 | 670 | 100 | 660 | 100 |
| **Ethnicity** | | | | | | |
| Asian or Asian British | 1,053,380 | 7.5 | 50 | 7.5 | 55 | 8.3 |
| Black, Black British, Caribbean or African | 488,714 | 3.5 | 23 | 3.5 | 30 | 4.5 |
| Mixed or Multiple ethnic groups | 230,442 | 1.6 | 11 | 1.6 | 20 | 3.0 |
| White | 11,926,307 | 84.9 | 569 | 84.9 | 548 | 83.0 |
| Other ethnic group | 353,232 | 2.5 | 17 | 2.5 | 6 | 0.9 |
| PNS | – | – | – | – | 1 | 0.2 |
| Total | 14,052,075 | 100 | 670 | 100 | 660 | 100 |

*NHS data for ages 20–29, not 18–29.

and actual figures from the recruited sample (18–19 year olds were included in the 20–29 group). Ages within the recruited sample ranged from 18–97 ($M$ = 54.0, $SD$ = 17.3) and were representative of NHS inpatients aged 18 to 80 years old, but were lacking in the > 80 groups. Sample demographics were representative in sex and ethnicity. See Appendix S2 in S1 File for more details of the current sample, such as education, marital status, and whether or not the patient had surgery.

## Hospital Stress Questionnaire

Total scores for the HSQ ranged from 67 to 609 out of maximum score of 670 ($M$ = 250.3, $SD$ = 112.6). The item rated as most stressful, on average, was "1. Not sleeping well", followed by "2. Feeling helpless or not in control" and "19. Missing loved ones" – see Appendix S3 in S1 File for the mean and standard deviation of each item, sorted from most to least stressful. The items rated as least stressful were "59. Not being able to pray or do other religious activities", "67. Not being able to smoke, drink alcohol, or use other substances", and "66. The hospital not meeting your individual needs (e.g., disability)". However, this is likely to be explained due to their lack of applicability to the majority of respondents rather than their salience to a few; a large proportion of respondents selecting 1 (not at all stressful) or N/A (scored in the analysis as 1), lowering the mean scores substantially.

## Exploratory factor analysis

The sampling adequacy was well above the acceptable standard (KMO = 0.97) and Bartlett's test of sphericity demonstrated that correlations between items were large enough to perform a factor analysis ($\chi^2$ (2211) = 28,724.73, $p < 0.001$). Considering the number of factors to retain, Kaiser's criterion and the scree plot suggested four factors, minimum average partial (MAP) test suggested six factors, and parallel analysis suggested eight factors. All three suggestions were examined: the four- and six-factor solutions had inadequate model fit statistics (CFI < 0.90), the eight-factor solution was also inadequate as only two items loaded onto the eighth factor without cross-loading. Therefore, a seven-factor solution was examined, which was deemed appropriate.

Within the seven-factor model, seven items did not load onto any of the factors (items #22, #23, #26, #30, #50, #55, #65). These items were excluded and the model was re-run, one item did not load onto any factors in this second iteration of the model (item #35), and so it too was excluded. After observing inter-item correlations, three items (#5, #20, and #29) correlated very highly ($r \geq 0.70$) with one or more other items, where each pertained to a similar stressor (e.g., listening and communicating, or rude and unfriendly), and so were considered to be repetitive, leading to a decision to exclude the redundant items. Finally, the model was run without the 11 above listed items, and in this last iteration of the model, only one item (#49) did not load onto any factor. Once this item was removed, all 55 remaining items loaded onto one of the seven factors, and the resulting model had acceptable fit statistics (CFI = 0.93, TLI = 0.90, RMSR = 0.03, RMSEA = 0.05), and accounted for 53.0% of the total variance. See Table 2 for the seven factors and their respective items, variance, and Cronbach's alpha values.

## Internal consistency reliability

For the resulting 55-item scale, a Cronbach's alpha value of 0.97 was obtained, this is above the accepted range of 0.70–0.95, likely due to the large number of items in the scale, which directly influences the value of alpha [41]. The scale has an average inter-item correlation of 0.36, which is within the acceptable range of 0.15–0.5 [42,43]. The corrected item-total correlation of each item was examined, where only one item (#67) had a correlation low enough ($r = 0.29$) to consider it for exclusion – recommendations of below 0.2 [44] or 0.3 [45] have been offered, since the correlation in question was on the upper end of these values, it was not excluded. Cronbach's alpha values for each factor were acceptable (0.73–0.91).

## Convergent validity

Convergent validity was assessed using Pearson's correlation coefficient. A large and significant correlation between the total scores of the PSS-10 and the final 55-item version of the HSQ ($r = 0.71$, $p < 0.001$) was found, suggesting that the HSQ is measuring the desired construct.

## Known-groups validity

Total scores on the HSQ-55 were compared between groups of participants. Mean differences were tested between those who had planned to stay in hospital (predicted to be subject to fewer stressors) (n = 223) versus those who were hospitalised in an emergency (n = 437). A t-test was performed ($t$ (438.3) = –5.89, $p < 0.001$), indicating that those with planned stays ($M = 179.1$) were significantly less stressed than patients with unplanned admissions ($M = 223.3$). Additionally, HSQ-55 scores were correlated with length of stay – a small, positive effect size was found ($r = 0.08$, $p = 0.049$), indicating that patients who stay in hospital longer

**Table 2.** Final seven-factor solution of the Hospital Stress Questionnaire (HSQ), containing 55 items.

| Factor | Items | Loading |
|---|---|---|
| **1. Quality of care** | | |
| Number of items = 11<br>9.4% variance explained<br>α = 0.91<br>Average inter-item correlation = 0.52<br>Test-retest: $r = 0.91$, $p < 0.001$ | 16. Feeling like the staff were not listening to you | 0.76 |
| | 7. The staff not being caring or friendly | 0.75 |
| | 52. Feeling like you were not being treated like a person | 0.51 |
| | 21. The staff not being responsive to the buzzer | 0.50 |
| | 13. The staff making a mistake that caused you harm | 0.49 |
| | 6. The staff being too busy | 0.46 |
| | 27. The hospital not being organised | 0.44 |
| | 9. The food being bad or not meeting your dietary requirements | 0.37 |
| | 25. Not feeling safe | 0.37 |
| | 8. Having to wait a lot | 0.36 |
| | 10. Feeling like you could not leave your bed or ward | 0.32 |
| **2. Away from home** | | |
| Number of items = 8<br>9.3% variance explained<br>α = 0.89<br>Average inter-item correlation = 0.50<br>Test-retest: $r = 0.85$, $p < 0.001$ | 32. Feeling homesick | 0.82 |
| | 19. Missing loved ones | 0.78 |
| | 18. Feeling lonely | 0.61 |
| | 43. Worrying about loved ones | 0.56 |
| | 41. Being in an unfamiliar place | 0.52 |
| | 12. Feeling bored | 0.40 |
| | 58. Missing your usual small comforts (e.g., hot tea) | 0.39 |
| | 54. Feeling like your life was on hold or you were missing out | 0.38 |
| **3. Inconvenienced** | | |
| Number of items = 11<br>7.3% variance explained<br>α = 0.87<br>Average inter-item correlation = 0.38<br>Test-retest: $r = 0.87$, $p < 0.001$ | 61. Not knowing the hospital rules | 0.63 |
| | 66. The hospital not meeting your individual needs (e.g., disability) | 0.52 |
| | 59. Not being able to pray or do other religious activities | 0.49 |
| | 60. Feeling like the staff focused on other patients more than you | 0.48 |
| | 62. Having to wear a hospital gown | 0.44 |
| | 47. Being reminded of loved ones who passed away while in hospital | 0.39 |
| | 31. Hearing or seeing emergencies | 0.37 |
| | 48. The staff not asking for consent before treating you | 0.37 |
| | 53. Worrying about money | 0.37 |
| | 67. Not being able to smoke, drink alcohol, or use other substances | 0.37 |
| | 44. Having to follow the hospital's schedule | 0.31 |
| **4. Health anxiety** | | |
| Number of items = 7<br>6.6% variance explained<br>α = 0.88<br>Average inter-item correlation = 0.51<br>Test-retest: $r = 0.79$, $p < 0.001$ | 15. Fearing your health will get worse | 0.67 |
| | 56. Not being sure of your diagnosis | 0.59 |
| | 11. Not knowing what was going to happen to you | 0.54 |
| | 39. Having to deal with the symptoms of your illness (e.g., sickness) | 0.47 |
| | 57. Worrying how you will cope once leaving hospital | 0.45 |
| | 2. Feeling helpless or not in control | 0.44 |
| | 14. Worrying that your treatment/medication will have side effects | 0.44 |
| **5. Negative effects of treatment** | | |
| Number of items = 5<br>4.9% variance explained<br>α = 0.73<br>Average inter-item correlation = 0.35<br>Test-retest: $r = 0.81$, $p < 0.001$ | 63. Needing help going to the bathroom | 0.60 |
| | 3. Having pain or discomfort from your treatment | 0.50 |
| | 46. Having tubes in your nose, mouth, or other body parts | 0.40 |
| | 24. Having to rely on others | 0.36 |
| | 64. Worrying that your appearance might change (e.g., scars) | 0.36 |

*(Continued)*

**Table 2.** (Continued)

| Factor | Items | Loading |
|---|---|---|
| **6. Ward environment** | | |
| Number of items = 8 | 4. Staying in a noisy room | 0.72 |
| 9.3% variance explained | 28. Sharing a room with strangers | 0.71 |
| α = 0.89 | 34. Being in an overcrowded ward | 0.64 |
| Average inter-item correlation = 0.49 | 17. The other patients being difficult | 0.58 |
| Test-retest: $r$ = 0.92, $p$ < 0.001 | 51. Feeling like you had no privacy | 0.58 |
| | 40. Being in a room that was too hot or too cold | 0.44 |
| | 45. Being in a room that was too bright or has no natural light | 0.37 |
| | 1. Not sleeping well | 0.36 |
| **7. Disrupted patient experience** | | |
| Number of items = 5 | 36. Medical procedure getting cancelled or delayed | 0.70 |
| 6.1% variance explained | 38. Equipment or supplies lacking | 0.69 |
| α = 0.84 | 37. Not being involved in the treatment plan | 0.50 |
| Average inter-item correlation = 0.51 | 42. Being in an unclean room | 0.42 |
| Test-retest: $r$ = 0.78, $p$ < 0.001 | 33. Not getting enough to drink | 0.34 |

are exposed to more stressors. However, we noted that this correlation was on the threshold of statistical significance.

## Predictive validity

A correlation was conducted between perceived in-hospital stress, using the HSQ-55, and both parts of the EQ-5D. The HSQ-55 asks questions pertaining to the in-hospital period ("during your hospital stay"), while the EQ-5D was adapted to pertain to the post-hospital period ("in the two weeks after being discharged from hospital"). A medium-sized, negative association was found between the HSQ-55 and both the EQ-5D-5L index values ($r$ = −0.35, $p$ < 0.001) and the EQ VAS ($r$ = −0.32, $p$ < 0.001), implying that as in-hospital stress increases, post-hospital health-related quality of life and self-rated health decrease.

## Test-retest reliability

The retest period was 14 days after the initial test. Test-retest reliability for the overall 55-item scale was excellent (ICC = 0.90, 95% CI = 0.81–0.95, $p$ < 0.001). For the seven individual factors, ICC values ranged from 0.77–0.91, where each correlation was statistically significant (all $p$ < 0.001), indicating "good" to "excellent" test-retest reliability.

## Medium-length version (28 items)

It was deemed necessary to produce shorter versions of the HSQ, as a 55-item questionnaire may be burdensome for particularly vulnerable groups (e.g., older adults with dementia or delirium, who make up a large proportion of NHS patients) or in surveys which already include several other measures. To produce a medium-length version of the HSQ, the four highest loading items within each factor were selected (see Table 3). Cronbach's alpha was good for the overall scale (α = 0.94), and for each of the seven factors (0.71–0.88). The HSQ-28 also showed good convergent validity ($r$ = 0.70), known-groups validity (planned ($M$ = 89.2), unplanned ($M$ = 113.2), $t$ (449.6) = −6.28, $p$ < 0.001), predictive validity (EQ-5D-5L: $r$ = −0.35; EQ VAS: $r$ = −0.32), and test-retest reliability (ICC = 0.87, 95% CI = 0.76–0.94).

**Table 3. HSQ-28: four highest loading items from each factor.**

| Factor | Items |
|---|---|
| **1. Quality of care**<br>α = 0.88 | 16. Feeling like the staff were not listening to you |
| | 7. The staff not being caring or friendly |
| | 52. Feeling like you were not being treated like a person |
| | 21. The staff not being responsive to the buzzer |
| **2. Away from home**<br>α = 0.84 | 32. Feeling homesick |
| | 19. Missing loved ones |
| | 18. Feeling lonely |
| | 43. Worrying about loved ones |
| **3. Inconvenienced**<br>α = 0.80 | 61. Not knowing the hospital rules |
| | 66. The hospital not meeting your individual needs (e.g., disability) |
| | 59. Not being able to pray or do other religious activities |
| | 60. Feeling like the staff focused on other patients more than you |
| **4. Health anxiety**<br>α = 0.82 | 15. Fearing your health will get worse |
| | 56. Not being sure of your diagnosis |
| | 11. Not knowing what was going to happen to you |
| | 39. Having to deal with the symptoms of your illness (e.g., sickness) |
| **5. Negative effects of treatment**<br>α = 0.71 | 63. Needing help going to the bathroom |
| | 3. Having pain or discomfort from your treatment |
| | 46. Having tubes in your nose, mouth, or other body parts |
| | 24. Having to rely on others |
| **6. Ward environment**<br>α = 0.84 | 4. Staying in a noisy room |
| | 28. Sharing a room with strangers |
| | 34. Being in an overcrowded ward |
| | 17. The other patients being difficult |
| **7. Disrupted patient experience**<br>α = 0.82 | 36. Medical procedure getting cancelled or delayed |
| | 38. Equipment or supplies lacking |
| | 37. Not being involved in the treatment plan |
| | 42. Being in an unclean room |

## Short version (10 items)

A short version was also produced by selecting the ten items of the HSQ-55 with the highest corrected item-total correlations (see [46,47]). Within these ten items, all seven factors were represented (one item from each of factors 2–7, and four items from factor 1; see Table 4). The HSQ-10 showed good internal consistency (α = 0.91), convergent validity ($r$ = 0.68), known-groups validity (planned ($M$ = 33.6), unplanned ($M$ = 44.2), $t$ (458.5) = −6.16, $p < 0.001$), predictive validity (EQ-5D-5L: $r$ = −0.32; EQ VAS: $r$ = −0.30), and test-retest reliability (ICC = 0.90, 95% CI = 0.80–0.95).

## 4. Discussion

The current study explored the psychometric properties of the HSQ, including reliability, validity, and factor structure. This was accomplished via a retrospective survey, completed by a diverse and representative sample of 660 recent inpatients. Results show that the measure is a valid and reliable tool for quantifying in-hospital psychological stress and is ready for use in future research

**Table 4. HSQ-10: items from the HSQ-55 with the highest corrected item-total correlation.**

| Items | Factor |
|---|---|
| 52. Not treated like a person | 1. Quality of care |
| 44. Having to follow hospital schedule | 3. Inconvenienced |
| 6. Staff busy | 1. Quality of care |
| 51. No privacy | 6. Ward environment |
| 27. Hospital unorganised | 1. Quality of care |
| 16. Staff not listening | 1. Quality of care |
| 11. Not knowing what will happen | 4. Health anxiety |
| 41. Unfamiliar place | 2. Away from home |
| 37. Not involved in treatment plan | 7. Disrupted patient experience |
| 24. Having to rely on others | 5. Negative effects of treatment |

– although it is yet to be tested on current inpatients. Item reduction, from an exploratory factor analysis, led to three versions of the HSQ being developed (55-item long version; 28-item medium version; 10-item short version), where each has been shown to have good psychometric properties.

The EFA also identified seven latent factors, each with an acceptable level of internal consistency, forming the subscales of the HSQ: quality of care, away from home, inconvenienced, health anxiety, negative effects of treatment, ward environment, and disrupted patient experience. Previous, similar in-hospital stress measures (HSRS [15], HSI [16], HRSQ-EP [17]) have produced comparable factors. Most, if not all, have a factor relating to the hospital environment (HSQ: ward environment), a change in lifestyle (HSQ: away from home), health-related fears (HSQ: health anxiety), difficulties with treatment (HSQ: negative effects of treatment), and a factor relating to the care received from hospital staff (HSQ: quality of care). The current measure introduced two new factors: being inconvenienced, and having a disrupted patient experience. The individual items and factors within the HSQ are capable of identifying clear areas of focus for improvement within the hospital setting, paving the way for future studies and policy decisions to improve the experience of hospitalisation.

Individual items were also examined, the highest scoring of which was "Not sleeping well". In a recent study, patients reported significant decline in sleep quality and quantity while in hospital, and these problems persisted for more than three months post-discharge [48]. Sleep disruptions in hospital such as early morning phlebotomy [49] and overnight NEWS observations [50] remain high, despite sleep loss in hospital being associated with cardio-metabolic derangements and an increased risk of delirium [51]. Other hospital-related stressors that were rated highly in the current study, such as pain ("Having pain or discomfort from your treatment") and noise ("Staying in a noisy room"), can exacerbate the problem of sleep disruption, leading to the risk of deleterious effects in immune function, wound healing, and mental health [52,53].

As seen above, Goldwater and colleagues [2] predicted problems with sleep and pain in their list of five hospital-related stressors: sleep disruptions, malnourishment and dehydration, mobility restriction, pain, and distressing environment and events. However, items in the HSQ that were designed to capture malnourishment (HSQ: "The food being bad or not meeting your dietary requirements"), dehydration (HSQ: "Not getting enough to drink"), and mobility restriction (HSQ: "Feeling like you could not leave your bed or ward") were reported as relatively low-ranking stressors in the current study. Goldwater's list is heavily weighted towards physiological stressors, and so is lacking in psychological (HSQ: "Feeling helpless or not in control") and environmental factors (HSQ: "Feeling like you had no privacy"). Such stressors comprised the majority of high-ranking items in the HSQ, and ought

to be acknowledged more in the post-hospital syndrome literature. For example, feelings of helplessness and not being in control was the second highest ranking item, and yet there is a paucity of research on this topic (see [54–56]).

It is clear that the hospital environment is acutely stressful, and aspects of the inpatient experience may not be conducive to a patient's recovery. Therefore, the authors recommend that interventions be designed to address the stressors rated most salient by patients in the current study, first and foremost of these is disrupted sleep, which leads to adverse effects both indirectly (via allostatic overload and post-hospital syndrome) and directly (see [57] for effects of sleep loss on recovery, and steps recommended to improve sleep in hospitals). Secondly, the HSQ may be used in future studies or clinical practice as an indicator of which patients may be most susceptible to the effects of post-hospital syndrome.

## Limitations

Several limitations were apparent within the current study. First, the representativeness of the sample is limited, due to the majority of the cohort being recruited via Prolific; a research participant platform. Notably, the current sample was more educated than the general public, and was lacking in participants aged over 80 years, a population which made up approximately 20% of NHS inpatients in 2022–23 (NHS Digital, 2023). Secondly, the retrospective nature of the current study raises concerns about the accuracy of the results, as it relies on the participant's ability to recall events from up to one year ago. Nevertheless, it is worth noting that issues relating to recall, if at play, are likely to have led to evidence of weaker psychometric properties than those observed here. Additionally, the predictive validity reported may be questioned by some, as the survey design was cross-sectional. Therefore, the authors suggest that future research addresses some of these limitations by employing the HSQ in hospitals to current inpatients, and adopting a longitudinal design to better assess the predictive validity of the measure with a battery of health outcomes.

## 5. Conclusion

The HSQ exhibits excellent reliability and validity, and an appropriate factor structure was identified. Three versions of the measure are available for use (short, medium, and long), each with acceptable psychometric properties. The HSQ may be used to justify and assess the effectiveness of interventions to reduce in-hospital stress, or to identify patients most at risk of experiencing post-hospital syndrome. However, the measure is yet to be tested on current inpatients, and further research is required to support the tool's ability to predict post-hospital adverse events and self-rated health.

## Supporting information

**S1 File. Appendices validation.**
(DOCX)

## Author contributions

**Conceptualization:** Daniel M. Ford, Rebecca Lawton, Elizabeth A. Teale, Daryl B. O'Connor.

**Data curation:** Daniel M. Ford.

**Formal analysis:** Daniel M. Ford.

**Funding acquisition:** Rebecca Lawton, Elizabeth A. Teale, Daryl B. O'Connor.

**Investigation:** Daniel M. Ford.

**Methodology:** Daniel M. Ford.

**Project administration:** Daniel M. Ford.

**Software:** Daniel M. Ford.

**Supervision:** Rebecca Lawton, Elizabeth A. Teale, Daryl B. O'Connor.

**Visualization:** Daniel M. Ford.

**Writing – original draft:** Daniel M. Ford.

**Writing – review & editing:** Rebecca Lawton, Elizabeth A. Teale, Daryl B. O'Connor.

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
