## [Decision Letter · Decision Letter 0]

3 Mar 2025

Psychometric validation of the Hospital Stress Questionnaire

PONE-D-24-59169

Dear Dr. Ford 

we are pleased to inform you that your manuscript has been judged scientifically suitable for publication and will be formally accepted for publication once it meets all outstanding technical requirements.

Kind regards,

Maria José Nogueira, Ph.D.

Academic Editor

PLOS ONE

1.Thank you for stating the following financial disclosure: [This research was funded by the National Institute for Health and Care Research (NIHR) Yorkshire and Humber Patient Safety Research Collaboration (NIHR Yorkshire and Humber PSRC) and the NIHR Yorkshire and Humber Applied Research Collaboration (NIHR Yorkshire and Humber ARC). The views expressed in this article are those of the authors and not necessarily those of the NIHR, or the Department of Health and Social Care.].

Please state what role the funders took in the study. If the funders had no role, please state: "The funders had no role in study design, data collection and analysis, decision to publish, or preparation of the manuscript.

Additional Editor Comments (optional):

Congratulations to the authors.

This work is robust and proposes a new instrument that is very relevant for the quality of healthcare to improve hospital environment.

Reviewers' comments:

Reviewer's Responses to Questions

**Comments to the Author**

1. Is the manuscript technically sound, and do the data support the conclusions?

Reviewer #1: Yes

2. Has the statistical analysis been performed appropriately and rigorously? 

Reviewer #1: Yes

3. Have the authors made all data underlying the findings in their manuscript fully available?

Reviewer #1: Yes

4. Is the manuscript presented in an intelligible fashion and written in standard English?

Reviewer #1: Yes

5. Review Comments to the Author

Reviewer #1: Congratulations to the authors for the relevance and significance of the work developed.

The path taken by the authors demonstrates methodological rigor in validating the instrument, which is highly relevant to the hospital environment, identifying the factors that contribute to stress.

Suggestions for Improvement:

In future work, it would be important to explore in greater depth the concepts involved in the presented topic, in order to better contextualize readers and strengthen the developed work, both in its more operational component and in its more practical aspect.

Methodological Quality:

The article demonstrates excellent methodological rigor, using appropriate samples and suitable statistical approaches for validating the instrument. The tests used are appropriate for assessing the instrument's validity. Furthermore, the sample used in the research is representative, increasing the robustness of the results.

References:

The references are extensive, up-to-date, and well-aligned with the subject matter, both in the more theoretical component and in the aspect related to the validity of the instrument.

6. PLOS authors have the option to publish the peer review history of their article (what does this mean? ). If published, this will include your full peer review and any attached files.

**Do you want your identity to be public for this peer review?** For information about this choice, including consent withdrawal, please see our Privacy Policy .

Reviewer #1: No

---

## [Editor Report · Acceptance letter]

PONE-D-24-59169

PLOS ONE

Dear Dr. Ford,

I'm pleased to inform you that your manuscript has been deemed suitable for publication in PLOS ONE. Congratulations! Your manuscript is now being handed over to our production team.

Kind regards,

on behalf of

Professor Maria José Nogueira

Academic Editor

PLOS ONE